# Dense Deployment of LoRa Networks: Expectations and Limits of Channel Activity Detection and Capture Effect for Radio Channel Access

**DOI:** 10.3390/s21030825

**Published:** 2021-01-26

**Authors:** Congduc Pham, Muhammad Ehsan

**Affiliations:** LIUPPA Laboratory, University of Pau, 64000 Pau, France; mehsantanoli@gmail.com

**Keywords:** LPWAN, LoRa, scalability, performance, channel access, carrier sense, channel activity detection, capture effect

## Abstract

With worldwide deployment of LoRa/LoRaWAN LPWAN networks in a large variety of applications, it is crucial to improve the robustness of LoRa channel access which is largely ALOHA-like to support environments with higher node density. This article presents extensive experiments on LoRa Channel Activity Detection and Capture Effect property in order to better understand how a competition-based channel access mechanisms can be optimized for LoRa LPWAN radio technology. In the light of these experimentation results, the contribution continues by identifying design guidelines for a channel access mechanism in LoRa and by proposing a channel access method with a lightweight collision avoidance mechanism that can operate without a reliable Clear Channel Assessment procedure. The proposed channel access mechanism has been implemented and preliminary tests show promising capabilities in increasing the Packet Delivery Rate in dense configurations.

## 1. Introduction

Low-Power Wide Area Networks (LPWAN) are an important component in the Internet-of-Things (IoT) maturation process. Behind this terminology are a variety of power efficient wireless communications over long distances. Technologies using ultra-narrow band modulation (UNB) such as SigFox or Chirp Spread Spectrum modulation (CSS) such as LoRa have become de facto standards in the IoT ecosystem [1] well before 5G standards. With large-scale deployment of such networks worldwide, it becomes critical to ensure scalability of such technologies, especially because most of them are currently deployed in the unlicensed bands.

### 1.1. Related Works

It has been long assumed that a LoRa network working in a given set of parameters (same frequency, same spreading factor, SF) is similar to a simple ALOHA system where performances dramatically drop when the number of nodes increases due to the lack of medium access control. Moreover, due to the extremely low throughput of these long-range technologies (in the range of 100 bps–30 kbps), the time-on-air (ToA) of a packet can be very large, typically in the range of several seconds, thus increasing further the probability of overlapping transmissions despite the limitation on the duty-cycle imposed by radio regulations in many countries.

When considering dense deployment scenarios, the first obvious question is “How does LoRa scale?”. This question has been addressed by a number of articles, [2,3,4,5,6,7,8] to name a few. All of these works clearly confirmed LoRa’s low performance when the number of nodes increases.

Early experiments and simulations of the capture effect (CE) phenomenon which offers the possibility for a packet to be decoded despite the presence of interfering nodes suggested that CE can somehow seamlessly improve the performance of LoRa networks despite an ALOHA-like channel access. Authors in Bor et al. [6], Croce et al. [9], Haxhibeqiri et al. [10] experimentally showed successful reception under concurrent transmissions conditions with LoRa’s modulation. Two important parameters were highlighted: collision start time and interfering signal strength. Their results can be summarized as follows: when the RSSI from the interfering signal is equal to or lower than the signal being interfered, and if the transmission of the interfering node begins after the first transmission’s, then the first transmission will be received correctly. They observed that synchronization with a transmitting node only needs correct reception of six symbols of the preamble at the receiver side. The authors in Croce et al. [9] confirmed that, when concurrent transmissions occur, the receiver might not be able to correctly demodulate the signal if the power of the interfering signal is significantly higher than the power of the reference signal. Experimentation with very accurate timing on capture effect has been realized in Rahmadhani et al. [11], and results confirmed those presented in Bor et al. [6] and Haxhibeqiri et al. [10]. The authors added an extension to a multiple gateway scenario where the reception signal strength can vary from one gateway to another.

One way to immediately increase scalability is to dynamically and optimally manage the allocation of resources such as SF for nodes instead of using the maximum SF12 value. The LoRaWAN specifications [12] (v1.1 at time of writing) already proposes a simple Adaptive Data Rate (ADR) mechanism for such purpose. However, authors in Li et al. [13] have studied the ADR efficiency and reported a long convergence time since many packets need to be processed by gateways prior to convergence. More sophisticated approaches have been proposed in [14,15,16,17,18,19,20,21] mostly to select SF parameters, and sometimes TX power as well. There are also some interesting works on scheduling LoRa transmissions using some variants of the TDMA approach when the network size is not too large [22,23,24] or optimizing modulations and coding schemes [25]. While a high SF value does not intrinsically increase the packet loss ratio, it does increase the probability of collision because of increased time-on-air. Therefore, most of the SF optimization propositions can definitely assign SF values to end-nodes in a more efficient manner to better exploit SF diversity.

However, even after an optimal SF allocation, nodes using the same SF will still be in the same collision domain. Therefore, the utilization of more advanced channel access methods operating with an underlying optimal SF allocation strategy has great potential to further improve the Packet Delivery Rate (PDR). There are fewer works discussing the issue of channel access methods for LoRa and to propose improvements. In recent works [26,27], authors have started investigating Carrier Sense Multiple Access (CSMA) (or so-called Listen-Before-Talk) approaches for LoRa networks. They mainly concluded that such CSMA approach can bring benefits in the form of increasing LoRa performances, compared to other types of channel access methods. In Pham et al. [27], the main contributions were notably to identify the issue of unreliable Clear Channel Assessment (CCA) when using LoRa’s CAD mechanism. A CSMA approach with an adapted random backoff procedure was therefore proposed for handling long packet transmissions under unreliable CAD conditions. Following this work, the authors in Ahsan et al. [28] studied three channel access protocols, LoRa-BED, LoRa-BEB, and LoRa-BEH, using LoRa’s CAD mechanism to sense the channel. They confirmed that exponential back-off (LoRa-BEB) works better in comparison to the other two protocols. In O’Kenned et al. [29], the authors also started to investigate the CSMA approach for the deployment of a wildlife monitoring infrastructure and concluded that a more elaborated channel access mechanism is needed when node density increases. A recent CSMA MAC protocol has been proposed in Gamage et al. [30], but it still mainly relies on CAD to detect channel activity. A variant proposes periodic beacons from the gateway, but this approach can only work with somehow synchronized nodes in addition to consume radio activity time at the gateway.

### 1.2. Contributions and Outline

This article presents extensive experiments on LoRa Channel Activity Detection (CAD) and Capture Effect (CE) property in a real-world deployment scenario to better understand how a competition-based channel access mechanisms at the MAC layer can be optimized for LoRa LPWAN radio technology.

We first present additional experimental tests on LoRa’s CAD mechanism to further describe the difficulties in achieving a reliable CCA procedure which is crucial in all CSMA approaches. Recent LoRa chips where the CAD mechanism is reported to be improved compared to previous generation LoRa chips have also been taken into account in this work. With the reliable CCA issue in mind, we then realized extensive experimental tests on a capture effect representative of a dense deployment scenario to state whether such property typical of frequency modulation approaches can be taken into account in the CSMA algorithm in order to overcome the unreliable CAD procedure of LoRa. At the light of these experimentation results, the contribution continues by identifying design guidelines for a channel access mechanism in LoRa and by proposing an original channel access method that can operate without CCA procedure. The article is therefore organized as follows. Section 2 first reviews the reliability issues with LoRa’s CAD and presents extensive experimental tests on CAD’s reliability, extending what we presented in Pham et al. [27]. Section 3 then presents experimental tests and results on LoRa’s capture effect in a dense deployment scenario. Based on these results, Section 4 reviews the main principles of distributed channel access in wireless networks, identifies and presents design guidelines for LoRa networks, and finally proposes and describes our new channel access mechanism. Discussions on reliability and efficiency, on energy considerations with an analysis of the additional energy consumption compared to the ALOHA (no control) approach, which is currently used in LoRa networks, and on the implementation and preliminary tests of the proposed channel access mechanisms are presented in Section 6. We conclude in Section 7.

## 2. Testing LoRa Channel Activity Detection

As a LoRa packet reception can be correctly realized below the noise floor, the use of the RSSI to assess for a clear radio channel is not reliable enough. For such a Clear Channel Assessment (CCA), LoRa chips embeds a dedicated Channel Activity Detection (CAD) procedure that usually takes a few LoRa symbols to operate. Figure 1 shows a LoRa PHY frame where a preamble is transmitted prior to the real payload transmission. The purpose of the preamble is to allow for the receiver’s radio to start synchronization with the transmitter’s radio. The header part (including the header CRC) is only used in variable length packet mode, which is, however, the most common mode in IoT LoRa networks. According to Semtech, the CAD procedure on the LoRa chip can efficiently detect the packet’s preamble. A “successful CAD” or “CAD returns true” means that radio activity has been detected (CCA is false) by the CAD procedure. “Unsuccessful CAD” or “CAD returns false” (CCA is true) means that no activity has been detected.

### 2.1. CAD Test Environment

The CAD tests are realized in the University of Pau (France) area. It is a typical terrain in the university with some low buildings and trees. A transmitter was placed on a tripod in a fixed position. LoRa data rate is initially set to SF12BW125 (spreading factor of 12 and bandwidth of 125 kHz), and the LoRa radio module was the SX1276. The choice of SF12 provides the longest time-on-air to test the CAD procedure. A CAD device continuously realizing a CAD procedure every 1000 ms and a simple receiver (denoted in the figure as gateway, but it is a more simple device than a gateway with a radio concentrator) were connected on a laptop computer as shown in Figure 2. The preamble length is set to the default value which is 8+4.25 symbols. Depending on the data rate, the preamble duration will therefore vary according to the symbol time. For SF12BW125, it is about 400 ms.

The transmitter node sends long packets of 244 byte (240 bytes payload + a 4-byte application-level header) every 15 s. The ToA of the packet is 8.69 s. The test setting is illustrated in Figure 3, and the experiment is referred to as “CAD experiment 1”, indicated by the blue square marked “1”.

### 2.2. CAD Experiment 1 Results

The transmitter is fixed at a given location. In Figure 3, it is indicated by the red arrow close to the blue square marked “1”. Then, we started moving the CAD device along the path illustrated in Figure 3. The environment along this path has low buildings and trees so it is typical of a mid-dense Non Line-of-Sight (LOS) environment. We observed a very fast decrease of the CAD procedure reliability when distance increases. In CAD experiment 1, for the first 400 m, the transmitter, the receiver, and the CAD device were in LOS conditions and the CAD procedure was stable and successful. It is worth noting that the CAD procedure is successful during the whole packet transmission, and not only for the packet’s preamble as presented by Semtech. Then, Figure 4 shows the CAD reliability results when the transmitter and the CAD device are separated by more than 400 m with some trees and small building in-between (non-LOS conditions). As can be seen here, the CAD procedure fails to detect channel activity many times during an on-going transmission.

In the test environment illustrated in Figure 3, for distances less than 1290 m in NLOS conditions, there have always been at least one successful CAD during an ongoing transmission. Then, from 1290 m and above, there were no successful CAD anymore while we were still capable to correctly receive the transmitted packet on the receiver device which is attached to the laptop, close to the CAD device (see Figure 2).

These tests confirmed the fact that, although a packet can be correctly received at several kilometers (even below noise floor with negative SNR), the CAD procedure can start to not reliably detect the whole transmission at a much smaller distance. We varied the SF—and the CAD interval accordingly—but SF didn’t seem to have any influence in this case and there were no visible changes. However, the experiment shows that, as opposed to description by Semtech, the CAD procedure on the SX1276 can also detect activity during the entire packet transmission, and not only during the packet’s preamble.

### 2.3. CAD Experiment 2 Description and Results

Some weeks after the first experiment, we received the new SX1262-based LoRa module. According to Semtech, the CAD procedure on the SX1262 has been optimized and can detect the whole packet transmission instead of only the packet’s preamble. As explained previously, this behavior was, however, already observed on the previous generation SX1276. An Application Note describing the tests and the performances of the new SX1262’s CAD procedure has been conducted by Semtech and reported in [31]. In addition, with the SX1262, the CAD procedure can be controlled by three parameters: cadDetPeak, cadDetMin and cadSymbolNum. According to [31], “the parameters cadDetPeak and cadDetMin define the sensitivity of the LoRa modem when trying to correlate to actual LoRa symbols. These two settings depend on the LoRa spreading factor and bandwidth, but also depend on the number of symbols (cadSymbolNum) used to validate or not the presence of signal”. These tests were, however, realized in a lab in a conducted mode where a signal generator was attached the SX1262 device under test. In this second round of experiments, we wanted to compare both SX1276 and SX1262 in a real long-range scenario to see if the new CAD procedure of the SX1262 can bring any improvement.

CAD experiment 2 uses the same setting (SF12BW125) for the fixed transmitter positioned as illustrated in Figure 3, the red arrow close to the blue square marked “2”. The transmitter position has been changed to be able to test along a longer path with less obstacles than for experiment 1. Two devices performing continuous CAD have been programmed, one with the previously used SX1276 and the other with the new SX1262. These two devices are fixed on a box put on a bike in order to be able to move faster and further on the bike lane. Each device has a red LED that is switch ON and OFF according to a radio activity detection or not. Here, the continuous CAD procedure is realized every 100 ms. cadDetPeak, cadDetMin and cadSymbolNum are set to their optimal value as recommended by Semtech in [31]: cadDetPeak=28, cadDetMin=10 and cadSymbolNum=4. The experiment setting is illustrated in Figure 5.

In CAD experiment 2, for the first 1.3 km, both CAD devices detect the ongoing transmission in a very stable way: the red LED was ON during the whole packet transmission. Between 1.33 km and 1.9 km, the CAD procedure varies from unstable—there have always been at least one successful CAD during a transmission, therefore a behavior similar to the one illustrated in Figure 4—to very unstable where some packet transmissions could not be detected at all by both devices. The main results of CAD experiment 2 is that we unfortunately did not observe any significant difference between the SX1276 and the new SX1262.

## 3. Capture Effect in LoRa

In most of the previous works on CE, in order to get accurate timing, the devices were placed close together and were all connected to a timing unit. The results on CE we present in this article were obtained from a real-world deployment scenario, more representative of dense deployment of LoRa devices.

### 3.1. First Capture Effect Experimentation Settings

Our first experimentation setting consists of a very simple scenario to better catch Capture Effect property in a real LoRa deployment. We start with two transmitters (transmitter A and transmitter B) and one receiver (acting as gateway) as depicted in Figure 6: transmitter A and transmitter B are at about 20 m distance from each other, and the gateway is placed approximately in the center of the two nodes. The first set of tests is performed in LOS conditions.

During the test, both of the transmitters have the same maximum output power of 14 dBm and the LoRa data rate is SF12BW125. The choice of SF12 here facilitates the synchronization of nodes to study the impact of concurrent transmissions overlapping the packet’s preamble. Both transmitters have the same payload, which is 240 bytes (resulting in 244 bytes transmitted with the 4-byte application-level header), which will remain constant throughout the experiment. The ToA of the packet is therefore similar to the CAD experiment and is 8.69 s. All of the communication took place in the 868 MHz frequency band. Again, the LoRa packet preamble configuration is the default one that consists of eight symbols resulting in a final preamble length of 8+4.25=12.25 symbols. The preamble duration is therefore 401.41 ms. Two types of gateways were used: one is an Arduino Nano (single-channel with regular radio module) as shown in Figure 6 and the other is a Raspberry PI with the RAK2245 multi-channel radio concentrator hat (based on Semtech’s SX1301 concentrator) running a simple util_pkt_logger program.

To synchronize the transmitter nodes, we connected both nodes via male-male cable. After being switched on, transmitter A sends a high signal on a pin and transmitter B synchronizes its clock with transmitter A. Once the nodes are synchronized, they start broadcasting a packet every 25 s. We switched on a LED on both transmitters at the beginning of a transmission to visually see if the nodes are transmitting at roughly the same time and that they have successfully been synchronized.

We realized multiple runs of 20 packet transmission cycles. We started the test with a delay of 100 ms between transmitter A and transmitter B: if $t_{A}$ is the transmission time at transmitter A, transmitter B will send its packet at $t_{B}=t_{A}+delay$. delay will be incremented to 200 ms, 300 ms, 400 ms, 500 ms, and 600 ms. Note that a delay of 400 ms or greater will avoid concurrent transmission during transmitter A packet’s preamble in the SF12BW125 data rate. We expect to receive one packet per packet transmission cycle that would show the efficiency of the capture effect.

#### 3.1.1. Results

For the first delay of 100 ms, 20 packets were sent and, out of those 20 packets, 18 packets were successfully decoded/received at the gateway. All of the packets were received from transmitter B (the interferer). For 200 ms delay, out of 20 packets, we were able to receive eight packets from transmitter A. For a delay of 300 ms, 8 out of 20 packets were received from transmitter A and 1 packet from transmitter B. For a delay of 400 ms and above, we can see that most of the packets sent by transmitter A have been correctly received. From one run to another, the results are quite similar. The number of packets received from transmitter B for delays of 200 ms, 300 ms, and 400 ms can slightly vary from 0 to 2 packets, thus always remaining very small.

#### 3.1.2. Summary

We noted that a majority of packets were lost or were unrecoverable when there was a large preamble overlap between the two nodes. In Figure 7, this is the case when transmitter B (the interferer) starts after 200 ms and 300 ms. However, in most cases received, packets are from a transmitter that starts first, i.e., transmitter A. Once there was no overlap on the preamble (the delay is greater or equal to 400 ms), most of the packets were received again from the transmitter which starts first. At 100 ms delay, we can see that the interfering node can actually overcome the first transmission. In this simple CE experiment involving only two transmitters, we can conclude that to maximize the probability of a packet reception from a transmitter, it is necessary to avoid as much as possible overlapping transmission during the packet’s preamble.

### 3.2. Second Capture Effect Experimentation Settings

In a second setting, we increase the number of interfering nodes with again several pre-defined delays from the beginning of the transmission at a master node. The nodes are placed in an 100 m × 100 m area, and the purpose here is to reproduce a real dense LoRa deployment scenario to verify whether the observed behavior with two nodes is still valid for a denser network. We further consider two cases.

Case A has one master node and eight slave nodes. The delays between slaves nodes and the master node are increasing by a 100 ms step as illustrated in Figure 8. The purpose of case A is to verify whether there are any packets received when there are many nodes interfering during the packet’s preamble.

In case B, we have one master node and four slave nodes that will transmit successively after the packet’s preamble, as illustrated in Figure 9, respectively, with a delay of 500 ms, 1000 ms, 1500 ms, and 2000 ms compared to the packet from the master.

In both cases, the packet payload is 100 bytes (resulting in 104 bytes transmitted with the 4-byte header). The ToA of the packet is now about 4 s. Again, each node sends 20 packets with 8 s between each packet. Several runs were realized for each case.

#### 3.2.1. Results

For case A, almost all packets are either lost or corrupted: the last packet at each packet transmission cycle can not be decoded neither. For case B, we observed some packets received at the gateway. However, from one run to another, the number of received packets can greatly vary: from 1 to 14 for instance. The source of the correctly received packets can also change in a random manner, but we noticed a majority of packets from either an interfering node at 500 ms or an interfering node at 1000 ms.

#### 3.2.2. Summary

These two additional experiments show the limit of the capture effect property in a real and dense scenario. In case A, as almost all packets are corrupted, we can see that even the packet from the last interfering node (the interferer with delay of 800 ms from master node’s transmission) can not be correctly decoded. Preliminary results from the simple 2-node scenario suggested that capture effect could be beneficial to the last interfering node, provided that it starts its packet transmission sufficiently early (e.g., delay of 100 ms from the previous transmission) to take over any preceding transmissions. Case A experiment invalidates this expected behavior. In case B, we can see that, even if there are no overlapping transmissions during a packet’s preamble, we are far from correctly receiving one packet per transmission cycle as there are many corrupted packets.

## 4. What Channel Access for LoRa?

Although promising, the capture effect is not sufficient to avoid a high collision rate in dense LoRa deployment scenarios. We saw that, when the number of concurrent transmissions is high, a capture effect showed no real efficiency to allow the reception of at least one packet out of all the concurrent transmissions. Therefore, performances of the network would be somehow similar to those of an ALOHA system and a more advanced channel access mechanism is definitely needed.

### 4.1. Review of Channel Access Principles

Competition-based channel access mechanisms such as CSMA variants are widely used in a number of networking technologies because of their simplicity: they do not require centralized coordination nor high signaling overheads. However, such CSMA approach heavily relies on a reliable Clear Channel Assessment (CCA) to detect an ongoing transmission in order to avoid interfering with that transmission. CCA is then usually coupled to a random exponential backoff procedure to avoid synchronization issues between competing nodes. For instance, Figure 10 shows the IEEE 802.11 (WiFi) CSMA approach which works as follows.
Before initiating a transmission, a node first senses the channel in an attempt to detect an ongoing transmission from other nodes;If radio activity has not been detected during a DCF inter-frame space (DIFS), the node can proceed with the transmission. This is illustrated by a green DIFS);If there are some radio activities meaning that the channel is busy, which is illustrated by a red DIFS, then the node will continuously sense the channel to detect for the end of the transmission.Once the channel becomes free, the node needs to observe a free channel for an additional DIFS duration. If this is the case, then the node will initiate a random backoff counter expressed in number of slot times in the range [0,W−1];This random backoff counter will be decreased as long as the channel is free. When a transmission is detected (a competing node started a transmission earlier), then this counter will be frozen. The node will then wait for the channel to become free again for at least a DIFS duration before resuming to decrease the backoff counter;When the random backoff counter reaches 0, the node can start its packet transmission;W is initially set to 1. W will be doubled for each retry (this is the exponential backoff) until it reaches a maximum value,Applying a random backoff counter when the channel becomes free is necessary because several nodes may be competing and are all waiting for the channel to be free.

In LoRa, as shown in the previous sections, the CAD unreliability in a real-world deployment scenario will impair the efficiency of traditional CSMA approaches. There is unfortunately no observable improvement of CAD reliability with the new SX1262 LoRa chip.

CAD unreliability is somehow similar to the so-called “hidden terminal” problem where a node can not detect an out-of-range transmission from another node because the radio signal is too weak. In LoRa, while an ongoing transmission can not be detected, a correct reception is, however, still possible because of the unique feature of LoRa to be able to decode a signal below the noise floor. The hidden terminal problem can usually be handled by adding a collision avoidance mechanism (CA) with an RTS/CTS exchange prior to the transmission of a data packet. For instance, CSMA/CA on IEEE 802.11 (WiFi) networks can enable this RTS/CTS mechanism between nodes, and the access point as illustrated in Figure 11. Nodes far from the transmitting node can receive the CTS packet from the base station and can therefore determine the busy period based on the data packet length that can be advertised in the CTS packet.

While a LoRa gateway can somehow act as an access point to implement such a RTS/CTS mechanism, it is not really possible for two reasons. The first reason is that LoRa IoT nodes can not always listen for incoming packets as they are mainly in sleep mode and only wake up to transmit an uplink packet. Therefore, they can not constantly check for the CTS packet. The second reason is related to sub-GHz regulation on a duty-cycle, therefore considering frequent RTS/CTS exchanges between end-nodes, and the gateway is definitely not possible as the gateway is also subject to duty-cycling.

It is also relevant to look at IEEE 802.15.4 that is closer to the IoT application domain. The underlying MAC layer proposes both beacon-enabled mode with slotted CSMA/CA and non-beacon-enabled mode with unslotted CSMA/CA channel access. Here, we are describing the latter mode illustrated in Figure 12 because the beacon-enabled mode works with a central coordinator and necessitate a much higher level of synchronization which is definitely not possible in LoRa IoT networks. The IEEE 802.15.4 non-beacon-enabled with unslotted CSMA/CA mode works as follows:Prior to any packet transmission attempt, a node first has to wait for a random number of backoff periods in the range $[0.2^{BE}-1$], where BE is initially set to 3;At the end of the backoff timer, if the channel is free, then the node can immediately start its packet transmission;If the channel is sensed busy, BE is increased and the node waits for an additional $[0.2^{BE}-1$] backoff periods. BE can be increased until it reaches a maximum value;

The CSMA procedure in IEEE 802.15.4 always forces a backoff timer prior to any packet transmission when compared to IEEE 802.11 CSMA. In addition, IEEE 802.15.4 does not implement a continuous sensing of the channel to detect when the ongoing transmission ends. The exponential backoff procedure in IEEE 802.15.4 simply increases BE—thus the backoff timer internal, from one attempt to another when trying to transmit the same packet. These differences between IEEE 802.15.4 and IEEE 802.11 have their reasons. First, the continuous sensing of the channel to detect when the ongoing transmission ends is very costly in terms of energy, especially when the time-on-air of a packet is long. This is the case for IEEE 802.15.4 as it usually transmits at 250 kbps while IEEE 802.11’s throughput is well beyonds 50 Mbps. As IEEE 802.15.4 radios are more dedicated to small battery-operated devices, the approach to simply increasing the backoff timer interval is far less energy consuming. The second reason is related to both node density and traffic load: IEEE 802.15.4 networks are expected to less denser and less busier than WiFi networks. In addition, as the IEEE 802.15.4 for Wireless Sensor Networks or IoT mainly runs under the mesh topology (i.e., P2P and without any central coordinator) with a shorter radio range (i.e., lower transmit power), the spatial reuse is therefore higher. This contributes also to reducing the traffic load in the network.

While the CCA procedure can not be fully efficient in LoRa networks for the same reasons stated previously, the simpler backoff timer strategy of IEEE 802.15.4 seems more adapted to LoRa device.

### 4.2. Design Guidelines for LoRa

Given the aforementioned issues, we can enumerate the following guidelines for the design of a channel access mechanism for LoRa networks. It is important to note that LoRa networks are not designed for delay-constrained communications. Therefore, the design guidelines enumerated below are not suitable for these types of communications.
As CAD is not reliable enough to detect all ongoing transmission, the proposed approach can not entirely rely on CCA but should also have a collision avoidance mechanism similar to IEEE 802.11 RTS/CTS;For the same reason, it is not necessary—nor desirable because of energy consumption considerations—to continuously detect for the end of a transmission;Because of (1) and (2), a complex backoff timer management procedure with a freeze & resume mechanism as in IEEE 802.11 CSMA is not really tractable;However, as node density can be high, an initial backoff procedure prior to packet transmission similar to IEEE 802.15.4 CSMA/CA can still help in improving the temporal distribution of competing transmitter nodes;Since nodes are not continuously in receive mode, and also because of duty-cycle limitations, a congestion avoidance mechanism as stated in (1) can not implement the full RTS/CTS exchange mechanism. As CTS depends on the correct reception of an RTS, the only control packet that is really needed is the RTS;In order to receive the RTS indicating a future data transmission, a node willing to transmit needs first to listen for a sufficiently long period for an RTS;With an RTS packet carrying only the expected size of future data packet, the correct reception of an RTS can enable a NAV mechanism similar to the one of IEEE 802.11 RTS/CTS;While the majority of transmitter nodes should start by listening for an RTS, a minority proportion of transmitter nodes should start by sending the RTS. Therefore, a node willing to transmit will first determine whether it will start listening for RTS or start sending the RTS;

### 4.3. Proposed Channel Access Mechanism for LoRa

Following the aforementioned guidelines, our proposed channel access mechanism for LoRa is illustrated in Figure 13. It does not rely on a reliable CCA procedure and will use a short RTS control packet prior to data transmission to implement a light collision avoidance mechanism. As stated previously, we are not considering delay-constrained communications; therefore, the proposed solution is not adapted to these types of communications. Our concerns are to increase the Packet Delivery Rate (PDR) when concurrent transmissions can not be reliably detected and to do so while keeping energy consumption low.

Although not shown in Figure 13, a node willing to transmit first performs a CCA (with LoRa’s CAD procedure) to try to detect radio activity. If an ongoing LoRa transmission is detected, the node simply waits for a random period of time and will retry. In all our CAD experiments, we did not observe a false positive where activity is detected while there is no activity in reality. Therefore, we can consider that, when LoRa’s CAD detects activity, it is quite reliable. Figure 13 only illustrates our channel access approach when there is no activity detected.

The proposed channel access approach operates in three phases and a node willing to transmit a data packet will go through these phases in a sequential manner. Phase 1 consists of only a listening period in order to listen for RTS packets. Phase 2 consists of sending the RTS packet immediately followed by another listening period. Prior to the RTS transmission, a random backoff timer is applied. In phase 3, the data packet is transmitted. Again, prior to the data packet transmission, a random backoff timer is applied. The random backoff timer is expressed in number of DIFS whose value is set to the duration of the LoRa packet’s preamble. DIFS therefore depends on the LoRa data rate in order to adapt the various timer according to the time-on-air of packet transmission.

A node willing to transmit a data packet will either start by listening for an RTS with probability 1−P (phase 1) or start directly by sending an RTS with probability *P* (phase 2). By choosing P with a small value such as 0.1, only 10% of transmitters will actually start by claiming for the radio medium. The other transmitters will then listen for the RTS. In Figure 13, node $D_{i}$ will start by listening while nodes $D_{j}$ and $D_{k}$ start by trying to send an RTS. Prior to the RTS transmission, a random backoff timer expressed in number of DIFS is applied in order to further separate transmitter nodes starting directly in phase 2. The number of DIFS is taken in range [0,W]. As DIFS is set to the packet’s preamble duration (which can be about 400 ms in data rate SF12BW125), W should not be too large and a value such as 7 can provide a sufficient transmitter separation property.

Phase 1 has a duration of at least W×DIFS+TOA(RTS) in order to listen for an RTS sent after the maximum backoff timer interval. TOA(RTS) is the time-on-air of an RTS packet which carries the payload length of the future data packet to enable the NAV mechanism. With data rate SF12BW125, the time-on-air of a short packet with size between 1 and 5 bytes can be rounded to about 830 ms. It is sufficient to carry a packet header and a 1-byte field to indicate the future data packet length. Therefore, taking W=7, phase 1 can have a duration of about 7×400 ms + 830 ms = 3.630 s.

If a node receives an RTS during phase 1, it will simply stop the whole procedure and will retry later. The earliest time at which it can try again can be estimated using the NAV mechanism as illustrated in Figure 14. Otherwise, if no RTS is received, then the node will start phase 2 where it will transmit its own RTS after a random backoff timer. Immediately after the transmission of the RTS, phase 2 continues with a listening period whose purpose is to receive RTS from nodes that have started with phase 1 and that have longer random backoff timer at the beginning of phase 2.

As can be seen from Figure 13, nodes that start directly at phase 2 have higher priority over nodes starting with phase 1. Remember that, by carefully setting probability *P*, we can adjust the proportion of higher priority nodes. Then, amongst nodes starting in phase 2, those with a longer backoff timer interval will have higher priority because the other nodes with a smaller backoff timer interval would already have switched to a listening period. This hierarchy is illustrated in Figure 14, where node $D_{k}$ is finally the node that can transmit its data packet.

## 5. Implementation and Preliminary Results

### 5.1. Implementation

We implemented the proposed solution in our LoRa framework [32]. Depending on the LoRa data rate, the duration of the DIFS is set to the duration of the default packet’s preamble. We set W=7 and the listening period duration is defined as W×DIFS+TOA(sizeof(RTS)). As stated previously, for data rate SF12BW125, the duration of listening period is about 3.6 s for instance. The RTS packet is five bytes long (four bytes for the application header and one byte to indicate the size of the future data packet). The listening period is implemented as a regular LoRa packet reception. As explained previously, a LoRa radio actually provides a ValidHeader interrupt and an RXDone interrupt when a correct packet is received. The LoRa packet’s header is managed at the radio level; it is not a user or application level header. This header is inserted when the radio module works in an Explicit Header mode which is naturally the most common mode for sending variable length packet. The header size is 20 LoRa symbol according to Semtech. Thus, upon reception of the ValidHeader interrupt, a node will wait to receive the RXDone interrupt within a small time interval. If the packet is an RTS, the RXDone interrupt should be received shortly (less than 200 ms) after the ValidHeader interrupt because an RTS carries only the future data packet length (1 byte) as payload. If no RXDone interrupt is received within the small time interval after the ValidHeader interrupt, then the node assumes that it has detected an ongoing transmission, i.e., a data packet, and not an RTS. It can then act accordingly, i.e., it can immediately differ its packet transmission by a random period of time, see Figure 15 bottom figure. As it is not possible to know the payload size of the ongoing transmission, the random period of time should be at least larger than the time-on-air of the maximum packet length. For instance, for a 255-byte packet at SF12BW125, it is about 9 s.

### 5.2. Preliminary Tests and Results

With this implementation, we run again the scenarios depicted by Figure 8 and Figure 9 with data rate SF12BW125. As the nodes are placed in an 100 m × 100 m area, the CAD procedure is reliable at this distance. Therefore, we deactivated the CAD procedure to force all nodes to not detect radio activity in order to go into the collision avoidance mechanism as described in Section 4.3. Each node sends 20 packets in total in each run. However, as opposed to the previous experiment without channel access mechanism, here, as the proposed channel access mechanism automatically defers the packet transmission, the transmission of two successive packets is separated by a sufficiently large time interval, typically 120 s for case A and 50 s for case B. Packet size is again 104 bytes so time-on-air is about 4 s. Probability *P* is set of 0.1. As stated previously, for data rate SF12BW125, the duration of phase 1 (the first listening period) is about 3.6 s. Therefore, the maximum duration of phase 2 is twice this value, i.e., 7.2 s. As can be seen in Figure 15, when a node receives an RTS, it knows that it can go for a NAV period of at least a listening period plus 7×DIFS plus TOA(sizeof(DATA)), where sizeof(DATA) will be provided by the RTS packet. In our scenario setting, it is 3.6 s + 2.8 s + 4 s = 10.4 s. As explained above, if a node detects the beginning of a data packet, it will immediately go for an NAV random period of time of at least the time-on-air of the maximum packet length. Here, for a 255-byte packet at SF12BW125, it is about 9 s.

We attached two LEDS per nodes that allow us to visually track the main stages of the channel access mechanism running at the nodes: (a) listening period (LED1 ON), (b) backoff to transmit RTS (LED2 blink fast), (c) receive RTS or ValidHeader from DATA packet (LED1 from ON to blink fast), (d) NAV period (LED1 and LED2 ON), and (e) backoff to transmit DATA packet (LED2 blink fast during backoff, then ON from beginning of transmission until the end of transmission, i.e., about 4 s).

For each case, we realized five runs (each run consists of 20 packets from each nodes). For both cases, and for all the runs, the number of nodes starting directly at phase 2 was always either 0 or 1. The node starting at phase 2 always succeeds in transmitting its packet as its RTS packet is received by all the other nodes. From nodes competing at phase 1, possibly after a NAV period from the previous cycle, we observed that our proposition succeeded in successively assigning a transmission slot for each of the nodes.

The preliminary tests with the very dense scenarios of Figure 8 and Figure 9, however, showed that, when all the nodes start with phase 1 (no node starts directly in phase 2), then it is necessary to increase *W* when these nodes continue in phase 2. We added this modification. If there is at least one node starting directly in phase 2, then this modification has no impact as those nodes starting in phase 2 would typically receive an RTS and would go into an NAV period.

In the preliminary tests, we were capable of obtaining a PDR between 91% and 96% depending on the run. In total, each run has 9×20=180 data packet transmissions for case A and 5×20=100 data packet transmissions for case B. With the real experiments, we were not able to determine accurately whether there have been concurrent transmissions of RTS or not since the transmission of an RTS is very short, and whether reception of RTS did benefit from the capture effect or not. However, we can obtain a PDR that is far higher compared to what has been observed in the initial experiments without the proposed channel access mechanism, where almost all packets were lost or corrupted. To evaluate the proposed approach in more details, in future works, we will build a simulation model to study the performance of the proposed approach in a larger variety of nodes and traffic density, to get detailed statistics on the number of RTS/DATA collisions, to determine the impact of varying parameters *W* and *P*, and link them to traffic densities.

## 6. Discussion

### 6.1. Channel Access and Similarity with Neighbor Discovery Protocols

As CCA can not be reliably stated, the proposed channel access mechanism borrows ideas from Neighbor Discovery (ND) protocols in wireless networks with active/inactive periods [33]. The main objective here is therefore to maximize the probability to have overlapping active periods between nodes, and there are similarities with Birthday [34], DiffCode/ADiffCode [35], SearchLight [36], Nihao [37], and Griassdi [38] to name a few. The last three have been recently proposed in the context of IEEE 802.15.4 or BLE networks. However, the main difference between ND protocols and channel access protocols resides in the fact that ND protocols try to discover neighbor nodes much less frequently than a channel access protocol. Therefore, most of the ND protocols for wireless networks run in a somehow synchronized manner (with start frames and slots) and can afford to send a larger number of signaling packets (beacons packets for instance) during a longer discovery period. While our channel access protocol also wants to maximize overlapping of listening period and the RTS transmission period, the context of LoRa networks in which it operates is completely different and calls for an adapted and low complexity approach.

The SMAC [39] protocol proposed for managing multiple node’s schedules in Wireless Sensor Networks can also somehow be considered in this category of ND protocols. SMAC’s objective is to synchronize activity periods between various nodes in a multi-hop wireless networks. It has mainly been tested with IEEE 802.15.4 radios. The similarity between our approach and SMAC resides in the listening period for SYNC packets prior to an attempt to send a SYNC packet to become the master. This behavior is similar to our listening period for the RTS packet. However, our objective is very different and, in our channel access proposition, we combined several listening periods with several backoff procedures to maximize the probability to discover nodes willing to transmit in the same time window. In addition, SMAC operating with IEEE 802.15.4 radios still requires a reliable CCA and implements a full SYNC/CTS exchange while our proposition precisely avoids any CCA procedure and avoids costly RTS/CTS exchanges that are not compatible with duty-cycling radio regulations.

### 6.2. Reliability and Efficiency

The proposed approach can not completely avoid collisions because of the lack of a reliable CCA. Therefore, collisions resulting in corrupted packets received at the gateway which subsequently means lost/discarded packets is still possible. As LoRa networks operate in the ISM sub-GHz frequency band, which is subject to tight regulation on duty-cycle, this confirmed that uplink data that request acknowledgment messages scheduled by the LoRaWAN network server and sent back by the gateway are rarely used. There is therefore no easy way at the device level to detect that something went wrong. Our approach lays at the MAC layer, so, in case such confirmation is needed by the final application, the device can request an acknowledgment using the available higher-layer reliability mechanism. Some LPWAN networks propose to use a duplication approach. For instance, nodes with Sigfox radio technology transmit three duplicate packets in the purpose of increasing the PDR. LoRa networks do not implement this behavior although the end-device can be programmed to do so. However, this approach is very costly in terms of energy as it will be discussed later on.

The main objective of our approach is to greatly decrease the probability of concurrent data transmission with the use of RTS packet and implicit priority. The probability of concurrent transmission of RTS packet is reduced with the introduction of probability *P* that determines whether a transmitter starts by listening or by sending RTS. Then, using a random backoff timer prior to sending an RTS will further reduce the probability of the concurrent transmission of RTS packets. Ultimately, the concurrent transmission of RTS packets can benefit from a capture effect as we expect that this number of concurrent transmissions will be very small after applying probability *P* followed by a random backoff timer.

If we look back at the scenarios depicted previously in Figure 8 (case A) and Figure 9 (case B), we can see that the proposed approach can nicely handle the concurrent transmissions. In case A where there are about nine nodes transmitting in a small time window, the proposed approach will first apply a filter with probability *P*. Therefore, with P=0.1, out of the nine nodes, there will be probably one or two nodes maximum that will start directly at phase 2. Assuming two nodes starting at phase 2, *j* and *k* for instance, these nodes will compete for the medium with the random backoff timer as illustrated in Figure 13. The node with the largest backoff timer will be able to send its packet, while the other one will go into a NAV phase. Here, it is node *k* as illustrated in Figure 14. All the other nodes will start with phase 1 and will move into the NAV phase to defer their transmission upon reception of an RTS packet—normally the one sent by node *j*.

For case B, it is highly possible that all nodes start with phase 1. As depicted in Figure 15, there are several random delay components that will maximize the probability of receiving an RTS within a listening period in order to move into corresponding NAV periods. These delay components are indicated with orange blocks. It must be noted that, when a node finally transmits a data packet, other nodes that are in their listening period can also receive the beginning of the LoRa packet, see Figure 15 bottom figure. For a packet reception, a LoRa radio actually provides a ValidHeader interrupt and an RXDone interrupt. As the ValidHeader interrupt is part of the LoRa packet reception RF chain, it is more reliable than a CAD procedure—recall that LoRa can receive and decode packets below the noise floor. Therefore, upon reception of the ValidHeader interrupt for a data packet (and not for an RTS), a node in its listening period can also differ its packet transmission by a random period of time, in a similar way than if an ongoing LoRa transmission has been detected.

In dense deployment scenarios, where the number of nodes dramatically increase, the probability of collisions will definitely increase, even with our proposed solution. One common way to adapt a backoff-based approach to increase traffic load is to increase the upper bound on the random backoff timer to better distribute competing nodes in the time line. For instance, in the IEEE 802.11 CSMA approach, *W* is doubled for each retry resulting in the so-called exponential backoff procedure. Similarly, IEEE 802.15.4 increases the BE exponent every time the channel is still sensed to be busy after the initial backoff timer. To be fully efficient, this solution requires either low-level acknowledgment or a reliable collision detection or a reliable CCA procedure. None of those can be easily achieved in LoRa. With our approach, there are, however, two possible additional mechanisms that can be considered in future works. In the first one, a node can simply count the number of NAV to decide whether its local *W* should be increased or not. For instance, it is possible to immediately increase *W* after a NAV(DATA). In the second mechanism, a node can also report the number of NAV in the data packet that will be eventually transmitted to the gateway. At the gateway, all the NAV statistics from all the nodes can be processed to determine a traffic density level that can be reported in future scheduled downlink packets. With such information, a node can further adapt the value of *W* or *P* to the traffic conditions. The efficiency of these mechanisms still need to be studied extensively with simulations because increasing *W* has an impact on the listening period duration as well, while increasing *P* may increase the probability of RTS collision.

### 6.3. Energy Considerations

According to Semtech’s LoRa chip datasheet, the supply current for a transmission is about 30 mA at 14 dBm. For a reception, it is about 10 mA and can go down to 5 mA with the new SX1262 LoRa chip. We can take 5 mA to minimize the cost of a packet reception.

If we consider the no channel access approach and a node sending a 30-byte packet at data rate SF12BW125 once every 10 min, then the mean consumption for a 1 h period is about (TOA(30)×30 mA × 6)/3600 s = 0.0823 mA with TOA(30)=1.646 s. This can be denoted as $E_{phase3}$. We consider that, after transmission, the node goes in deep sleep mode where the consumption is very low (in the order of uA).

The maximum additional energy cost to transmit a data packet with our proposed channel access mechanism is $E_{phase1}+E_{phase2}$, when considering a node starting with phase 1. As stated previously, for data rate SF12BW125, the duration of phase 1 is about 3.6 s therefore $E_{phase1}=3.6\times$ 5 mA = 18 mA. For phase 2, we need to only add the cost for transmitting the RTS, assuming that, during the random backoff timer, the node can be put in deep sleep mode. Therefore, $E_{phase2}=E_{phase1}+TOA\left(RTS\right)\times$ 30 mA = 42.9 mA with TOA(RTS)=0.830 s. The maximum duration of phase 2 with significant power consumption is 3.6 s + 0.830 s = 4.43 s. Finally, the mean consumption for a one hour period is about (18 mA + 42.9 mA)× 6/3600 s = 0.101 mA, which should be compared to the 0.0823 mA with a no channel access mechanism. Expressed in terms of days of autonomy when operated on 2500 mAH batteries, the no channel access strategy can provide an autonomy of about 1265 days while our proposed channel access mechanism can run the node for about 1031 days. If we further consider that, out of the six transmission attempts per hour, 90% start with phase 1 while 10% start directly with phase 2 (assuming P=0.1), then the mean consumption for a 1 h period is now about ((18 mA + 42.9 mA) × 5.4 + 42.9 mA × 0.6)/3600 s = 0.098 mA, resulting in an increased autonomy to reach 1062 days.

It is interesting to compare with a duplication approach that is taken by some LPWAN networks. For instance, nodes with Sigfox radio technology transmit three duplicate packets to increase the PDR. With such duplication strategy, the mean consumption for a 1 h period for the no channel access approach is about 3×0.0823 mA = 0.246 mA, which is more than twice the maximum consumption of our proposed channel access mechanism without any guarantees that PDR can be increased by duplicating transmissions, especially in a dense scenario. Expressed in terms of autonomy, the 3-duplicate strategy can power a node for only 423 days!

## 7. Conclusions and Future Works

This article presented extensive experiments on the LoRa Channel Activity Detection and Capture Effect property in order to better understand how a competition-based channel access mechanisms at the MAC layer can be optimized for LoRa LPWAN radio technology. We found that the CAD procedure is not reliable enough to provide an efficient Clear Channel Assessment procedure that is at the foundation of CSMA channel access methods. The Capture Effect, which seems promising to allow for correct reception of packets even under overlapping transmissions, shows its limits when several nodes are involved. It can therefore not compensate for the weakness of the CAD procedure. In light of these results, we enumerated some design guidelines for a channel access mechanism in LoRa radio and proposed a channel access method with a lightweight collision avoidance mechanism that can operate without a reliable Clear Channel Assessment procedure. By taking an approach that has some similarities with Neighbor Discovery protocols, the proposed channel access method can greatly reduce the probability of concurrent transmissions while inducing only a small additional overhead in energy consumption. The proposed channel access mechanism has been implemented and preliminary tests show promising capabilities in increasing the PDR. In future works, we will further analyze the performances of the proposed approach with larger scale simulations to determine the impact of varying the operational parameters such as *W* and *P*, and link them to traffic densities.

## Figures and Tables

**Figure 1 sensors-21-00825-f001:**
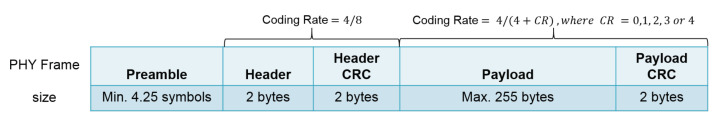
LoRa PHY frame format.

**Figure 2 sensors-21-00825-f002:**
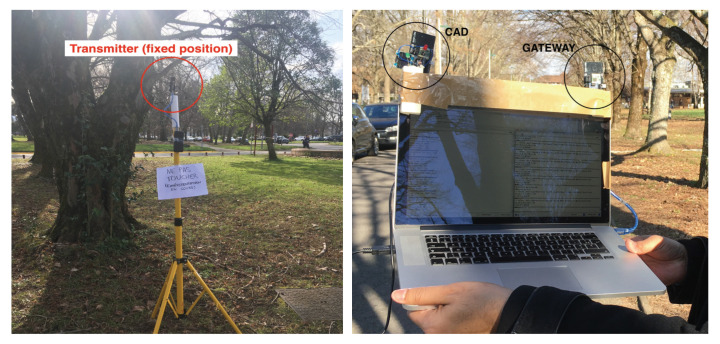
CAD Transmitter and gateway.

**Figure 3 sensors-21-00825-f003:**
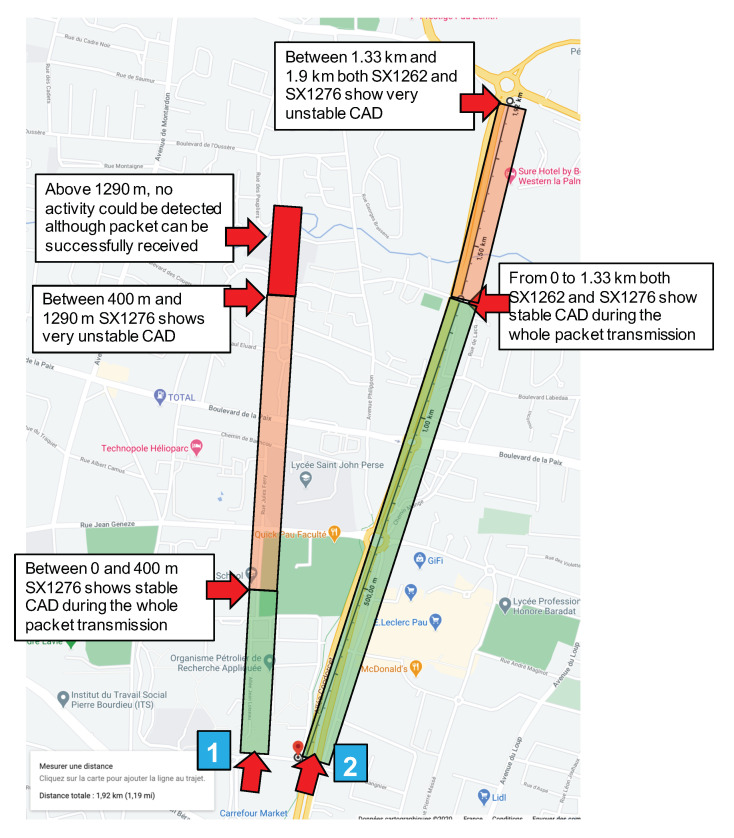
CAD test environment map.

**Figure 4 sensors-21-00825-f004:**
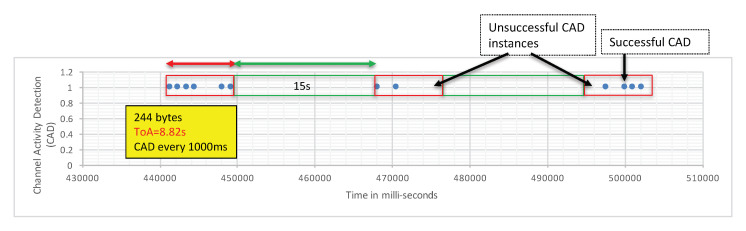
CAD fails to detect activity of on-going transmissions.

**Figure 5 sensors-21-00825-f005:**
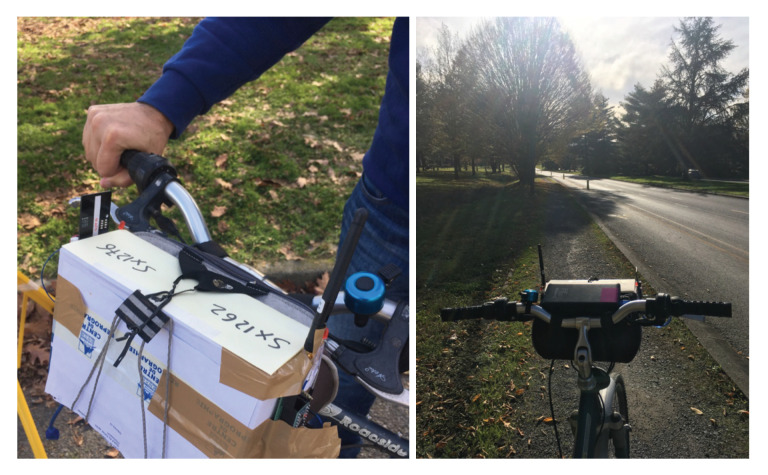
Comparing CAD performance on SX1262 and SX1276.

**Figure 6 sensors-21-00825-f006:**
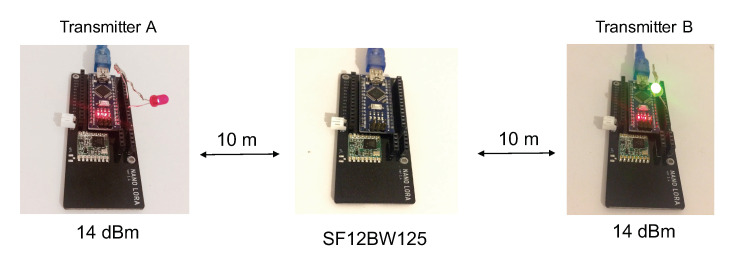
First CE experimentation setting.

**Figure 7 sensors-21-00825-f007:**
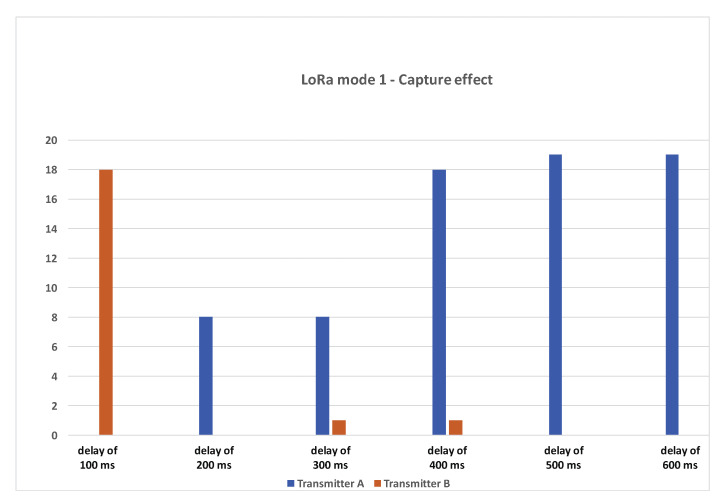
Capture effect test results.

**Figure 8 sensors-21-00825-f008:**
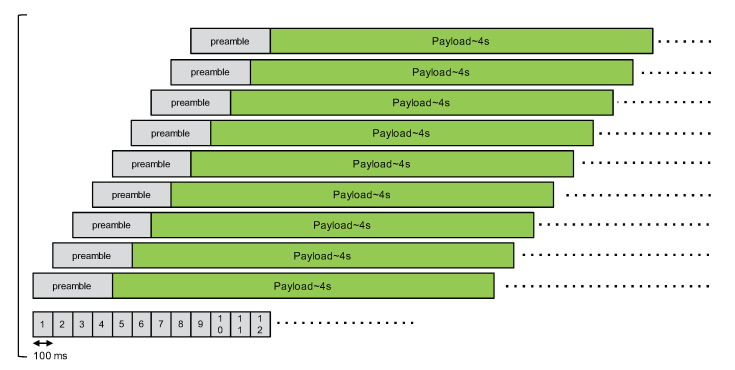
Second CE experimentation, case A.

**Figure 9 sensors-21-00825-f009:**
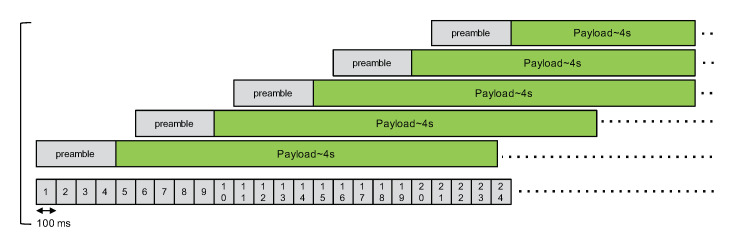
Second CE experimentation, case B.

**Figure 10 sensors-21-00825-f010:**
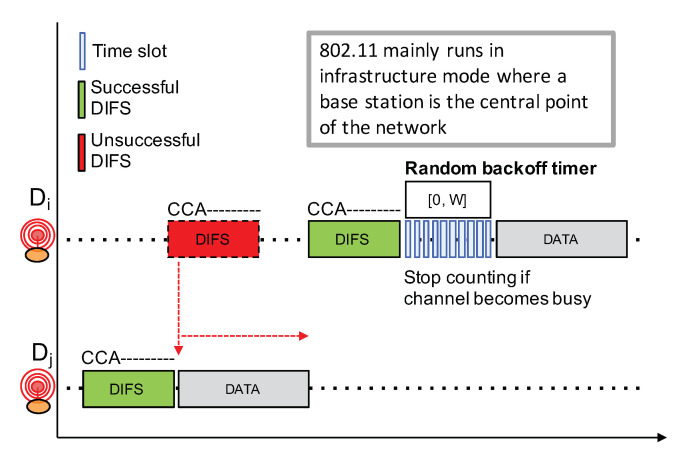
IEEE 802.11 CSMA.

**Figure 11 sensors-21-00825-f011:**
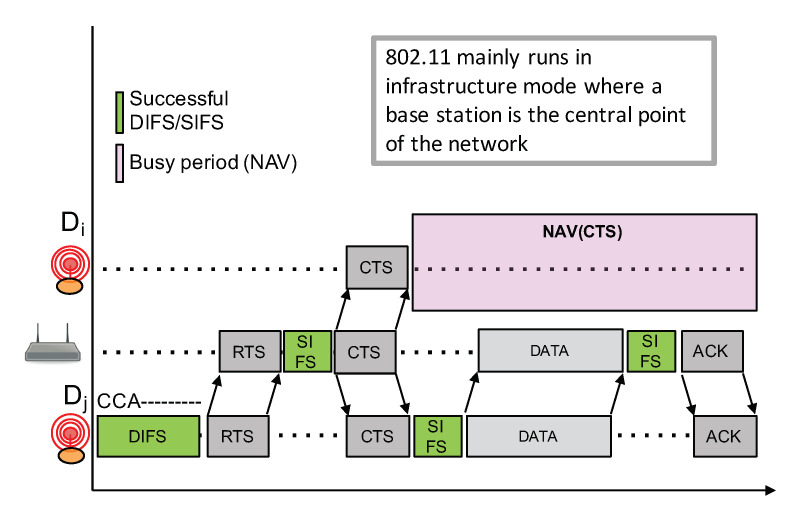
IEEE 802.11 CSMA/CA RTS/CTS.

**Figure 12 sensors-21-00825-f012:**
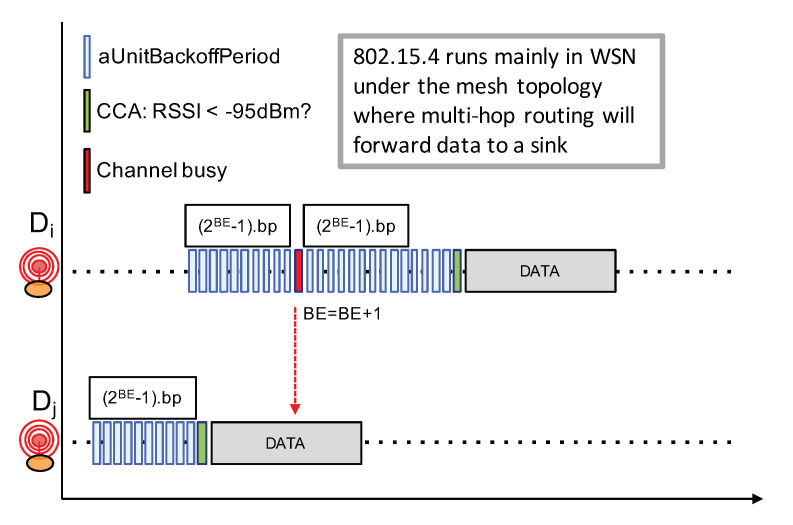
IEEE 802.15.4 non-beacon unslotted CSMA.

**Figure 13 sensors-21-00825-f013:**
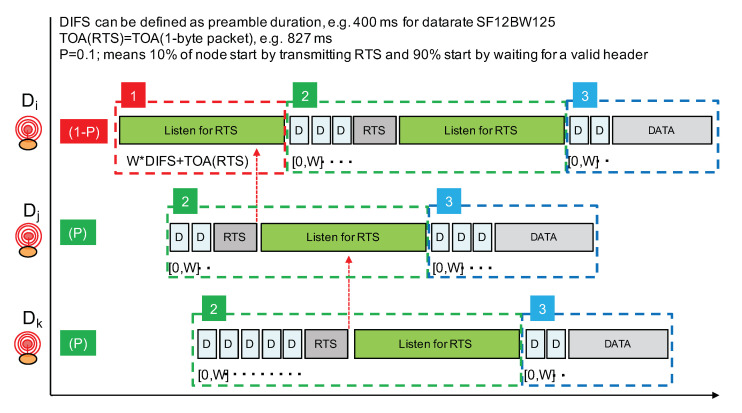
Principle of a channel access mechanism without CCA for LoRa.

**Figure 14 sensors-21-00825-f014:**
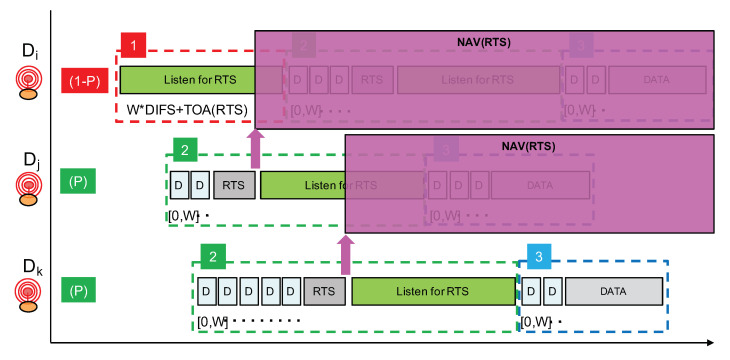
Using a network allocation vector.

**Figure 15 sensors-21-00825-f015:**
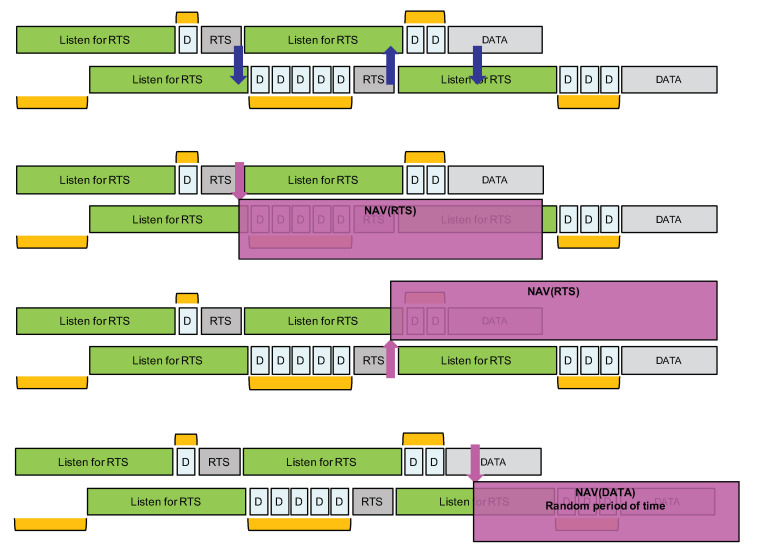
Case B: maximize overlapping of Listen/RTS periods.

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
