# Peer review of "Dense Deployment of LoRa Networks: Expectations and Limits of Channel Activity Detection and Capture Effect for Radio Channel Access"

_sensors, 2021, doi:10.3390/s21030825_

Round 1

Reviewer 1 Report

In the paper “Dense Deployment of LoRa Networks: Expectations and Limits of Channel Activity Detection and Capture Effect for Radio Channel Access”, a new channel access method with a lightweight collision avoidance mechanism is proposed to ensure the operation without a reliable Clear Channel Assessment procedure. The paper is laid out logically and is clear and easy to follow. To the best of my knowledge, the research is novel and original. However,there are some comments I would offer for consideration:

(1) Although the proposed method can sort the transmission, it still does not rule out all the transmission error. What should I do with the wrong transmission result? Is there a way to fix it or retransmit again?

(2) The proposed method decreases the probability of concurrent data transmission with the use of RTS packet and implicit priority. However, it cannot completely avoid collisions. As the number of transmitters increases, this defect will become more obvious. Is there a way to deal with this situation, for example, treat all transmitters whose RTS come at the same time as invalid and order them to re-enter the sorting queue.

Reviewer 2 Report

Very interesting reserarch in LoRa LPWAN systems, but is necessary to do a lot of changues.

1.- Introduction is very short. Is necessaty to review the state of art in more specific way. We have a lot of papers about LoRa.

2.- Why SF of 12 is very unestable. You can find information about select of SF in: https://doi.org/10.1016/j.iot.2019.100121 or https://doi.org/10.1016/j.comnet.2020.107491 or http://dx.doi.org/10.3390/en13030517 please check and incorporate a correct SF or make an study with differents LoRa parameters (BW, CR and SF).

3.- Paper is not well organized, relative works are in point 4.4, result section are no clear

4.- is necessary to define clearly the test done

5.- A little number of references are used. Is necessary to do a great changes

Major revision

Round 2

Reviewer 2 Report

Author made changes. But i have question.

1.- Paper is not well organized. By exaple section 5.5 must be go to begin o section 5, before see result we must to know the test done, no then that expose results.

2.- I agree with authors that SF 12 have a longst time on air. But produces a high packet losses ratio. I provide different papers to check, please change. If the authors dont want to change SF is necessary to provide a comparitive of loss packet information in relation with others SFs.
